# A Principled Approach to Natural Language Watermarking

## ABSTRACT

Recently, there is a surge in machine-generated natural language content being misused by unauthorized parties. Watermarking is a well-recognized technique to address the issue by tracing the provenance of the text. However, we found most existing watermarking systems for texts are subject to ad hoc design and thus suffer from fundamental vulnerabilities.

We propose a principled design for text watermarking based on a theoretical information-hiding framework. The watermarking party and attacker play a rate-distortion-constrained capacity game to achieve the maximum rate of reliable transmission, i.e., watermark capacity. The capacity can be expressed by the mutual information between the encoding and the attacker's corrupted text, indicating how many watermark bits are effectively conveyed under distortion constraints. The system is realized by a learning-based framework with mutual information neural estimators. In the framework, we adopt the assumption of an omniscient attacker and let the watermarking party pit against the attacker who is fully aware of the watermarking strategy. The watermarking party thus achieves higher robustness against removal attacks. We further show that the incorporation of side information substantially enhances the efficacy and robustness of the watermarking system. Experimental results have shown the superiority of our watermarking system compared to the state-of-the-art in terms of capacity, robustness, and preserving text semantics.

## CCS CONCEPTS

• **Applied computing** → **Text editing**; • **Security and privacy** → **Social aspects of security and privacy**; • **Computing methodologies** → *Natural language generation.*

## KEYWORDS

Watermarking, Natural Language Processing

**ACM Reference Format:**
. 2024. A Principled Approach to Natural Language Watermarking. In *Proceedings of (MM'24)*. ACM, New York, NY, USA, 14 pages. https://doi.org/10.1145/nnnnnnn.nnnnnnn

## 1 INTRODUCTION

Unauthorized distribution of texts is an ever-present threat, which even worsens with the rapid development of language models. Not only may articles written by human be stolen for misuse, but texts generated by machine could also be appropriated. For example,

ChatGPT may be abused for spreading fake information [9] or plagiarism of financial report [21]. With it comes the growing concern about inspecting the source of the texts for intellectual property protection or detection of language model misuse. Thus tracing text provenance to establish the ownership of text contents has become an urgent issue.

Digital watermarking is an effective approach to provenance tracing, and has been widely applied to images [6, 28], audios [13], and texts [1, 12, 23, 25–27]. By hiding a piece of information (i.e., watermark) into digital carriers and later recovering it, watermarking provides ownership evidence of the carriers. Particularly for texts, it is required to embed bit strings while maintaining good text utility. Rule-based methods [23, 25] are mostly founded on synonym substitution which is powerful in preserving utility, but neglecting the context often results in semantic changes. Context-aware methods [26, 27] improve the semantic consistency between the original and watermarked texts, yet merely robust against simple substitution attacks [27]. More recently for machine-generated texts, Abdelnabi and Fritz [1] propose an adversarial watermarking transformer to learn watermark embedding without ground truth-word substitutions and their locations. Kirchenbauer et al. [12] propose a watermarking scheme for detection of texts generated by large language models (LLMs), but it cannot trace identity at a finer granularity.

As we observe, most existing designs for text watermarking are ad hoc, lacking grounded theoretical basis and thus are prevented from wider deployment. First, the watermarking *capacity*, suggesting how long the bit sequence can be encoded into the text to be accurately decoded, mostly remains vague in the literature. The works of Abdelnabi and Fritz [1], Yang et al. [26], Yoo et al. [27] can only embed a fixed-length bit sequence, which eliminates any possibility to extend ID sequences for distinguishing more identities. Second, most watermarking schemes consider *weak adversaries*. For Yang et al. [26], Yoo et al. [27], naive word insertion, deletion, and substitution attacks are only taken into account. For Abdelnabi and Fritz [1], it assumes a black-box attacker who cannot access the watermarking model. Such an assumption raises the third common issue: *security by obscurity*, i.e., the security mechanism relies on hiding the details, or some parameters of the design — which is widely rejected by the community. It is easy for the adversary to follow the substitution rule and use the public infill model to detect and remove the watermark embedded by Yang et al. [26], Yoo et al. [27]. Likewise for Abdelnabi and Fritz [1], its security depends much on the secrecy of the watermarking models, which explains its poor performance under white-box attacks.

To address the above issues, we resort to the theoretical basis of the information-hiding framework [19] to seek a principled design for the text watermarking system. The key idea is to view watermarking from a communication system perspective — the watermarking capacity is formulated as the maximum rate of reliable transmission between the encoder and the decoder. Meanwhile, the rate should be achieved in the presence of a distortion-constrained

attack channel where the attack strategy is adjusted according to the watermarking strategy. Hence the capacity is the value of a mutual-information game between the watermarking party and the attacker. Our system is robust against realistic attacks in a transparent manner rather than relying on 'security by obscurity', as the attacker has complete knowledge about the watermarking approach, enabling it to derive the best attack strategy.

However, the theoretical framework is ideal given Nash equilibria and saddle points rarely exist in the capacity game. Nevertheless, we follow the approach of Moulin and O'Sullivan [19] to extend the information-hiding framework with distortion constraints and side information. Importantly, the side information, remaining hidden from the adversary, plays a key role in improving both capacity and robustness of our system.

To realize such a theoretical framework, we adopt a learning-empowered approach inspired by the recent progress in semantic communication. Rather than embedding bits into exact symbols transmitted, our system interprets the semantics of the texts and paraphrases the texts to embed watermarks. Our system is composed of an encoder-decoder net and an attacker net, learning to maximize and minimize the transmission rate, respectively, under the constraints that the watermarked text should not be deviating too much from the original one. We optimize the neural estimators to estimate mutual information which fundamentally exhibits the capacity and robustness requirements while the distortion constraint preserves the semantics of the text.

Our contributions are summarized as follows. First, we propose the first principled design for natural language watermarking based on the theoretical information-hiding framework. Second, we build a semantic learning-based watermarking system to enable end-to-end training for the rate-distortion-constrained capacity game between the watermarking party and the attacker. We show each component of our design, in particular, the mutual information loss, the side information, etc., all contributes to an improved capacity and robustness of our system. Last, we verify the advantage of our principled design by comparing it against multiple state-of-the-art watermarking systems. Experimental results demonstrate the superiority of our method in capacity, robustness, and preserving the semantics of the texts.

## 2 RELATED WORK

Natural language watermarking embeds a bit string into a text by altering it, and later extraction of this string provides evidence of text ownership. Our work is closely related to the following works.

Moulin and O'Sullivan [19] provide an information-theoretic analysis for information-hiding systems. The work formalizes hiding capacity which upper-bounds the rates of reliable transmission and quantifies the fundamental tradeoff between achievable information-hiding rates and distortion levels. Our work instantiates the theoretical framework for the first time via text-semantic learning. Through the theoretical lens, we observe most existing works for watermarking subject to ad hoc designs, violating fundamental principles. A detailed discussion can be found in Sec. B.

Traditional natural language watermarking methods are mostly **rule-based**. Some methods adjust the appearance of fonts, spacing, etc. throughout the document [4], vulnerable to attacks such as copy-paste, OCR, and re-typing. Some methods employ lexical substitution, replacing words with synonyms selected from dictionaries [5, 23, 25], ignoring the context and leading to potentially unnatural watermarked text due to inappropriate synonym choices. Other methods modify sentence structures, making substantial alterations to the original texts [16, 22]. However, these changes are not universally applicable thereby limiting their capacity.

To avoid the drawback of rule-based methods in generating rigid watermarking patterns, Yang et al. [26] proposed **context-aware** lexical substitution which dynamically generates synonym candidates based on the text context learned by masked language model BERT [7]. However, the scheme does not consider robustness: an attacker can remove up to 80% of the bits of the watermark by running the same process in CALS and fill the watermark positions with other words, reported by the authors.

To improve the robustness, Yoo et al. [27] proposed a watermarking scheme that considers potential text corruption on watermarked text. It leverages named entity recognition, unsupervised method YAKE and computation of NLI entailment score to identify features that are semantically or syntactically essential, and uses them as anchor points for watermarks. A corruption-resistant infill model is also trained to be robust on possible types of corruption including word insertion, substitution and deletion. However, the work only considers weak adversaries, and the watermark capacity is limited by the replaceable positions determined by the keywords.

Different from synonym substitution, some methods embed watermarks by **paraphrasing** the text conditioned on bit strings. For these watermarking systems, a key feature is the encoder-decoder framework that learns how to embed and extract watermarks. Adversarial watermarking transformer [1] belongs to this category. It has a pair of transformers as the encoder and decoder, which are trained to minimize the bit error of the predicted watermark. Meanwhile, it uses a discriminator to enhance the naturalness of the watermarked text. However, the robustness performance of their work is unsatisfactory in face of white-box attacks. Since our watermarking system also requires an interpretation of the text semantics, our work falls into this category, but we take a principled approach by formulating the information-hiding game between watermarking party and the adversary.

In some works, the watermark is no longer a string of bits, but only a statement that the text is watermarked or not. Kirchenbauer et al. [12] proposed such a method as a post-processing procedure of texts generated by LLMs. Before a word is generated, it selects a randomized set of 'green' tokens and promotes the use of green tokens during sampling. The method is only applicable to LLMs, aiming to protect the intellectual property of language models.

## 3 PROBLEM FORMULATION

**User cases.** We consider watermarking as a defense against both intellectual property (IP) theft as well as LLM abuse.
- *IP Protection*: Authors embed watermarks (unique identifiers) in their writings to prove ownership if being copied.
- *Provenance Tracking*: Responsible LLM owners watermark generated texts before release to later identify potential misuse (e.g., tracking fake news).

The watermarking system could serve as a white-box model for personal use or as an API service. In the first scenario, users train the system and use it to detect watermarks in texts crawled online (e.g., IP protection). The watermark here works as an alarm to notify users of suspicious texts. In the second scenario, watermark embedding and extraction are presented as API services for any authorized party to track the provenance of the text.

**Dilemma between public verification and robustness to removal attacks**: it has been recognized that if a watermarking system allows *anyone* to verify watermarks, the system would also allow attackers to iteratively alter the text until the watermark is removed, thus failing to defend watermark removal attacks [14]. It is a design choice of our work to sacrifice public verifiability but only to provide service to authorized parties, while achieving robustness against removal attacks.

The **requirements** of watermarking are as follows:
◇ *High-capacity.* The watermark should not only be accurately embedded into and extracted from the system, but also be sufficiently long to exhibit a diverse patterns to prevent mimicking attacks.
◇ *Transparency.* The watermarked texts should be natural to read without affecting the original meaning of the texts. Moreover, the watermark pattern should be stealthy to avoid being detected by the attacker, which facilitates the removal of the watermark.
◇ *Robustness.* In synergy with the transparency requirement, the watermarking system should resist a variety of removal attacks. Ideally, if an attacker is able to successfully remove the watermark, it has to alter the text to an extent that destroys its readability or even changes the original meaning of the text.

**A theoretical framework.** Our watermarking system follows the conventional information-hiding paradigm proposed by Moulin and O'Sullivan [19] (Fig. 1). We denote the host data (original text) as $S \in \mathcal{S}$, side information as $K \in \mathcal{K}$, and the watermark bitstring (message) as $m \in \mathcal{M}$. The encoder function $f$ creates watermarked data $X$ from $S$, $K$, and $m$. This $X$ is then attacked by a channel $A(y|x)$ that tries to remove $m$. The decoder $\phi$ uses $Y$ (the attacked data) and $K$ to estimate the watermark $\hat{m}$.

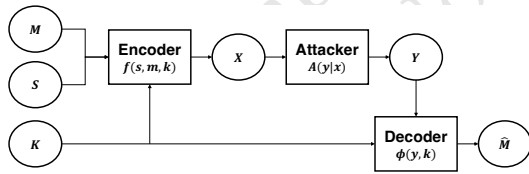

**Figure 1: The information-hiding framework.**

The watermarking system should resist removal attacks. The party controlling $f$ and $\phi$ aims to accurately recover the watermark despite attacks on the watermarked data. In contrast, the attacker tries to hinder this recovery. We assume that the attacker is omniscient that it knows the watermarking strategies and takes moves accordingly. Hence the adversarial game is formulated as:

$$\max_{f,\phi} \min_{A} J(f, A, \phi). \tag{1}$$

The transparency requirement is formulated as a constraint, shown in Eq. (2). The attacker also does not intend to alter the text

significantly, thereby limiting the distortion constraint between $X$ and $Y$, shown in Eq. (3).

$$D_{S,X} \triangleq \sum_{s \in S} \sum_{k \in K} \sum_{m \in M} p(s,k)d_1(s, f(s, m, k)) \leq D_1, \tag{2}$$

$$D_{X,Y} \triangleq \sum_{x \in X} \sum_{y \in Y} d_2(x, y)A(y|x)p(x) \leq D_2. \tag{3}$$

## 4 A PRINCIPLED WATERMARKING SYSTEM

In this section, we provide our principled design to the natural language watermarking system.

### 4.1 System Design

Following the requirement in Sec. 3, we propose a learning-based watermarking system inspired by the recent progress in semantic communication for text transmission.

Taking text $s$, watermark bit string $m$, and side information $k$ as inputs, the *encoder* outputs a watermarked text $x$ within the distortion constraint of Eq. (2). The main body of the encoder contains a transformer encoder which maps $s$ into the feature space and a transformer decoder that composes the features of $s$, $m$, and $k$ to construct $x$. The *attacker* attempts to erase the watermark from $x$ by feeding it into a transformer with a similar structure to the encoder, and outputs the de-watermarked text $y$, which is also known as the corrupted watermarked text. The attacker is also constrained by Eq. (3) as it expects to keep the semantics and readability of $y$. Given $y$, the decoder extracts a bit string $\hat{m}$ which is an estimate of $m$, by a transformer encoder concatenated with a linear layer mapping features of $y$ to the feature of $\hat{m}$.

It is important to note that the side information $k$ is shared between the encoder and the decoder, but not available to the attacker. The possible side information ranges from an empty message to any data dependent or independent of $s$. Examples of side information can be: the original text, a hash value of the original text, partial information of $s$, locations of the watermark, a random string, etc. As discussed in Moulin and O'Sullivan [19], without the side information, it is likely for the omniscient attacker to decode the watermark and to remove it from $x$. From a game-theoretic perspective, such side information gives the watermarking party an advantage to win and poses as a key component to robustness. Meanwhile, the existence of $k$ incapacitates public verification since only those who have $k$ could extract the watermarks. Our system can also be public verifiable with $k$ being none, yet losing some robustness according to the dilemma arguments in Sec. 3.

To enhance the transparency, a discriminator, modeled as a transformer encoder, is set to tell $s$ and $x$ apart by producing a score between 0 and 1, indicating the resemblance of $x$ to $s$. Such a score can be propagated backward to the encoder to keep the generated texts resembling the original. To enhance the discriminative capability of the discriminator and consequently improve the robustness of the watermarked $x$, we incorporate the watermark bit string $m$ as a condition to the discriminator, resulting in a structure akin to a conditional GAN [18]. The encoder plays as the generator fighting against the discriminator by minimizing the discriminator loss $L_{\text{disc}}(X, S|M)$, *i.e.,* drawing the distribution of $x$ closer to that of $s$.

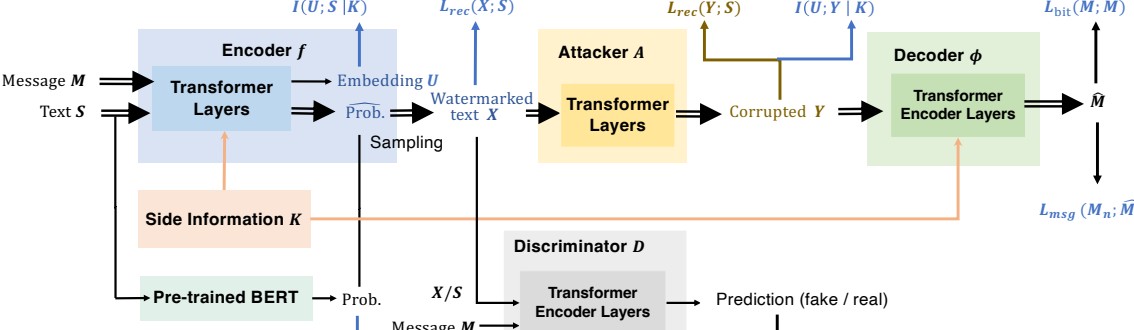

**Figure 2: Overview of our principled watermarking system.**

In constructing the framework, we inherit the setting of Moulin and O'Sullivan [19] that the attacker knows the encoder and decoder models, watermarked text but not the side information or the original text. This is in accord with the omniscient attacker assumption that the watermarking party can only assume the worst-case adversary in designing its strategy, and the fact that the attacker cannot access original texts or the side information.

## 4.2 Training Losses

The encoder-decoder net of the watermarking party is trained under the omniscient attacker assumption. We follow Thm. 4.4 of Moulin and O'Sullivan [19] to give the rate of reliable transmission for the watermarking party. By the information-hiding paradigm, we use $U$ to denote the encoding of the encoder, by which the transformer decoder constructs $X$. The payoff function consists of mutual information losses that

$$J(f, A, \phi) = I(U; Y|K) - I(U; S|K), \qquad (4)$$

where $I(U; Y|K)$ represents the mutual information between encoding and text corrupted by the attacker, $I(U; S|K)$ denotes the mutual information between encoding and the original text, both conditioned on the side information.

Following the dirty paper theorem (Eq. (7.7) in El Gamal and Kim [8]), the watermarking party aims to maximize the first term to recover the watermark from $Y$, while minimizing the second term to restrain the impact of the original text on $U$. The attacker's payoff is negative to that of the watermarking party to undermine the hiding capacity. The attack strategy is chosen from the class of $\{A(f, \phi)\}$ with distortion constraints $D_1, D_2$. Hence the problem of seeking the optimal watermarking strategy turns to be a rate-distortion-constrained capacity game. It is pointed out by Moulin and O'Sullivan [19] that the optimal attack is the solution to a rate-distortion problem and the optimal watermarking strategy is the solution to a constrained capacity problem.

**Mutual information neural estimation.** It is difficult to write the exact expression for the mutual information. Fortunately, mutual information neural estimation (MINE [3]) provides a theoretical

lower bound $I_\Theta(X; Y)$ for the approximation of $I(X; Y)$. Specifically,

$$I_\Theta(X; Y) = \sup_{\theta \in \Theta} \mathbb{E}_{P_{(X,Y)}}[T(\theta)] - \log(\mathbb{E}_{P_X P_Y}[e^{T(\theta)}]) \qquad (5)$$

$$\triangleq \sup_{\theta \in \Theta} L_T(X; Y) \qquad (6)$$

where $T \in T_\Theta$ is a network with parameters $\theta \in \Theta$, $P_{(X,Y)}$ denotes the joint probability distribution, and $P_X, P_Y$ means the marginal probability distribution of $X$ and $Y$, respectively. Hence $I(X; Y)$ can be estimated by maximizing $L_T(X; Y)$ on sampled instances from corresponding distributions over $\theta$. Here we adopt a transformer encoder concatenated with a linear layer as the model for $T$.

In the case where side information $K$ is present, for example, in calculating $I(X; Y|K)$, we sample from the joint distribution $P_{(X,Y,K)}$, and the marginal distributions $P_X$ and $P_{(Y,K)}$.

**Message reconstruction.** To successfully embed a bit sequence $M$ as watermark into the original text, the watermarking party trains its pair of encoder and decoder with a bit-by-bit message reconstruction loss $L_{bit}$. We compute $L_{bit}$ as the binary cross entropy loss between the embedded message $M$ and the predicted one $\hat{M} \in \{\hat{M}^x, \hat{M}^y\}$, where $\hat{M}^x$ and $\hat{M}^y$ respectively denotes the watermark decoded from the watermarked text $X$ and the corrupted text $Y$. We also provide the decoder another label $M_n$ for every $n$-bit sequence embedded to distinguish watermark patterns at the message level. For example, a 1-out-of-16 categorical label is assigned to a 4-bit watermark. Combined, the message reconstruction losses are formulated as

$$L_{bit}(M, \hat{M}) = \text{BinaryCrossEntropy}(M, \hat{M}),$$
$$L_{msg}(M_n, \hat{M}_n) = \text{CrossEntropy}(M_n, \hat{M}_n). \qquad (7)$$

**Distortion constraints.** In terms of Eq. (2) and (3), different forms of $d_1, d_2$ can be applied as the distortion constraints. We propose two distance measures: the first is the text reconstruction loss and the second is a discriminator loss. The former reconstructs $S$ from $X$ or $Y$ by the cross entropy loss, denoted by $L_{rec}(\cdot, \cdot)$. The discriminator loss is computed as a part of the conditional generative adversarial network (GAN) structure to distinguish $X$ from $S$ conditioned on message $M$ which is embedded in $X$. That is,

$$L_{disc}(X, S|M) = -\log(Disc(S, M)) - \log(1 - Disc(X, M)) \qquad (8)$$

where $Disc$ means the discriminator network. The network seems to discriminate $X$ and $S$, it in fact infers the presence of $M$ in the given texts. The encoder $f$, on the other hand, pits against the discriminator to generate $X$ close to $S$.

We denote the distortion constraints as $L_{D_{S,X}}$ and $L_{D_{X,Y}}$:

$$L_{D_{S,X}} = L_{\text{rec}}(S,X) + \max(0, -L_{\text{disc}}(X,S|M) - D_1), \quad (9)$$

$$L_{D_{X,Y}} = L_{\text{rec}}(X,Y). \quad (10)$$

The negative of $L_{\text{disc}}$ is an equivalent representation of the Jensen-Shannon (JS) divergence between $X$ and $S$ at convergence [10]. The encoder gets penalized when the negative of $L_{\text{disc}}$ goes over threshold $D_1$. In practice, we use $L_{\text{rec}}(Y,S)$ to replace $L_{\text{rec}}(X,Y)$ in Eq. (10) for better and stabler optimization performance. It does not break our assumption on the omniscient attacker by informing it about the original text.

**Knowledge distillation.** To promote the naturalness of watermarked text, we integrate a knowledge distillation module to let the encoder learn language characteristics from a pre-trained BERT model [7]. Specifically, we minimize the following cross-entropy loss between the output of the encoder and the probabilistic prediction of BERT provided the same text context. This knowledge distillation can also be viewed as a 'soft' reconstruction in which synonyms are used to construct the watermarked text. We carefully exclude the punctuations or special tokens as they do not have suitable substitutes. Thus the knowledge distillation loss is:

$$L_{\text{kd}} = -\sum_{i:s_i \notin \mathcal{P}} \sum_{k=1}^{|\mathcal{V}|} P_{\text{bert}}(s_i = k|S_{-i}) \cdot \log P_{enc}(x_i = k|S,m)$$
$$-\sum_{i:s_i \in \mathcal{P}} \log P_{enc}(x_i = s_i|S,m) \quad (11)$$

where $\mathcal{V}$ is the vocabulary set, $P_{\text{bert}}(s_i|S_{-i})$ is the probability of the $i$-th word in the original text $S$ predicted by BERT, and $P_{enc}(x_i|S,m)$ is the probability of the $i$-th word in the watermarked text $X$ predicted by the encoder.

To sum up, the watermarking party optimizes the payoff function $J(f,A,\phi)$ and the message reconstruction loss while keeping the distortion distance $D_{S,X}$ under control, by jointly updating the encoder and decoder:

$$L_{\text{wm}} = -J(f,A,\phi) + L_{\text{bit}} + L_{\text{msg}} + L_{D_{S,X}} + L_{\text{kd}}. \quad (12)$$

The attacker, plays against the encoder and decoder by optimizing the payoff in the opposite direction. However, we observe that the mutual information loss as the attacker's payoff does not deliver performant attacks, and thus we adopt a simpler but more powerful attack in practice:

$$L_{\text{atk}} = L_{\text{rec}}(Y,S), \quad (13)$$

referred to as 'adaptive attack' since the attacker can 'adapt' to any the watermarking strategy by learning to recover the original text from the watermarked text directly. Notice that we have weight factors for each additive loss term above and omit them here for conciseness.

## 4.3 Two Stages of Watermarking

Our watermarking system works at two stages: an offline stage for training different components of the system and an online stage for the actual watermarking encoding and decoding service.

---

**Algorithm 1** Training of the principled watermarking system.

---

**Input:** (a) Text set $\mathcal{S}$, side information generator $\mathcal{K}$; (b) encoder $f$, decoder $\phi$, attacker $A$, discriminator $Disc$, MINE net $T_1, T_2$; (c) distortion constraints $D_1, D_2$; (d) MINE iterations $N_m$, discriminator iterations $N_d$, attacker iterations $N_a$, epochs $N_e$

**Output:** Trained encoder $f^*$, trained decoder $\phi^*$

1: **for** $t_e = 1$ to $N_e$ **do**
2:     **for** $s$ in $\mathcal{S}$ **do**
3:         $k \leftarrow \mathcal{K}(s)$, $m \leftarrow$ random bits, $x \leftarrow f(s,m,k)$, $y \leftarrow A(x)$, $\hat{m} \leftarrow \phi(y,k)$;
4:         **for** $t_m = 1$ to $N_m$ **do**
5:             Calculate $L_{T_1}(U;Y,K)$; Calculate $L_{T_2}(U;S,K)$;
6:             Update $T_1, T_2$ by maximizing $L_{T_1}, L_{T_2}$ respectively;
7:         **end for**
8:         **for** $t_d = 1$ to $N_d$ **do**
9:             Calculate $L_{\text{disc}}$ by Eq. (8);
10:             Update $Disc$ by minimizing $L_{\text{disc}}$;
11:         **end for**
12:         **for** $t_a = 1$ to $N_a$ **do**
13:             Calculate $L_{\text{atk}}$ by Eq. (13);
14:             Update $A$ by minimizing $L_{\text{atk}}$;
15:         **end for**
16:         Calculate $L_{\text{wm}}$ by Eq. (12);
17:         Update $f$ and $\phi$ jointly by minimizing $L_{\text{wm}}$ ;
18:     **end for**
19: **end for**
20: $f^* \leftarrow f$; $\phi^* \leftarrow \phi$;

---

**Offline stage.** The training procedure of our framework is shown in Alg. 1. In each training iteration, we follow the workflow in Fig. 1 to optimize $f$ and $\phi$, with additional inner loops optimizing the mutual information estimators $T_1, T_2$, the discriminator, and the in-loop attacker channel.

**Online stage**: The watermarking system runs as open APIs including an encoder and a decoder. The user (text owner) first registers its identity-representing watermark with the system, and submits its created text, and side information (if any) to the encoder API. The API returns watermarked text. For the case without side information, anyone can verify ownership by submitting the target text to the decoder API to extract the watermark. For the case with side information, the authorized verifier submits the key along with the target text to the decoder API for watermark extraction. Finally, our system decides whether the extracted watermark agrees with the previously registered one to confirm the ownership of the target text.

## 5 EVALUATION

We evaluate our watermarking system on real-world natural language datasets from the capacity, transparency and robustness perspectives by comparing it with the state-of-the-art baselines. Ablation study is performed on the payoff function, which reveals the necessity of principled design. Then we investigate how different forms of side information affect watermarking. We also show how our framework can be applied to ownership verification. Due to space limit, we leave more experimental results to Appendix C, D

and G, w.r.t. the capability of watermark removal attacks, the ablation study, and the transferability of our system to unseen datasets, respectively.[1]

## 5.1 Experimental Setup

**Datasets.** We use the word-level WikiText-2 (WT2 [17]) dataset for evaluation. The dataset is curated from Wikipedia articles, containing about 2 million words for training, 200k words for validation and 200k words for testing. The training set of WT2 is a publicly available and unwatermarked dataset, which is accessible to both the watermarking party and the attacker. It is used for offline training as well as training the attacker model. The validation and test sets are considered private unwatermarked datasets that are inaccessible to the attacker. They are used to assess the performance of watermarking techniques.

**Baselines.** We compare our method to AWT [1], CALS [26] and IF [27]. AWT is the pioneering work that applies transformers to watermarking, following a machine-translation framework. CALS and IF are the two latest works that utilize context-aware rules to locate the embedding positions, and employ infilling models to select candidate words for watermarking. They are both featured by excellent naturalness, readability and fluency.

**Metrics.** To evaluate the performance of watermarks, we use the following metrics according to the requirements in Sec. 3.

• *Capacity* of a watermarking scheme refers to the maximum number of bits that can be embedded in each token, i.e., bit per token (BPT). Considering that the encoding-decoding process may introduce errors, we also adopt bit accuracy as a performance indicator, which is the ratio of the number of correctly decoded bits to the total number of bits. For fairness, we assess message bit accuracy across different BPTs to constitute a holistic view of capacity.

• *Transparency* is gauged by the ability to preserve the original meaning of the sentence. We use meteor [2] and SBERT [20] scores between the original and watermarked sentences for transparency. Meteor score evaluates the quality of machine translation output from combined perspectives of precision, recall, stemming, and synonymy. SBERT score measures the l2 norm between features of two sequences, where features are the output of a pre-trained BERT base model. Higher meteor scores and lower SBERT distances indicate better semantic similarity and relatedness.

• *Robustness* refers to the system's ability to survive watermark removal attacks. To evaluate the *robust capacity* of a watermarking system, we measure the message bit accuracy under various removal attacks at different BPTs. In addition, we measure the difference in transparency metrics before and after attacks to demonstrate the degradation of text semantics. A larger difference indicates that the attacker must compromise significant text utility in order to eliminate the watermark.

**Implementation details.** The default setting of our experiments is as follows. We select a dimension size of 512 for all transformer blocks and embeddings. Each transformer encoder or transformer decoder contains 3 layers with 4 attention heads. The feedforward dimension is set to 2048. The dropout rate is set to 0.1. Both $N_d$ and $N_a$ in Alg. 1 are set to 1, and the training batch size is set to 50. We use the Adam optimizer [11] with $\beta_1 = 0.9$, $\beta_2 = 0.98$

---

[1]The appendix is attached in supplemental materials.

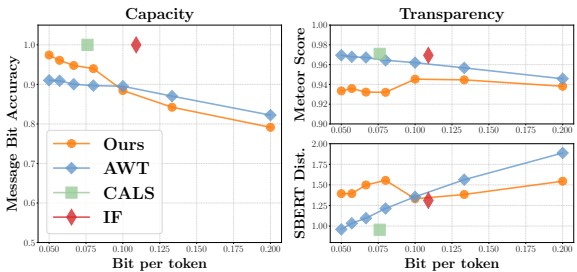

**Figure 3: The capacity and transparency performance of different watermarking schemes. (Our setup: $D_1 = 0$, K=None)**

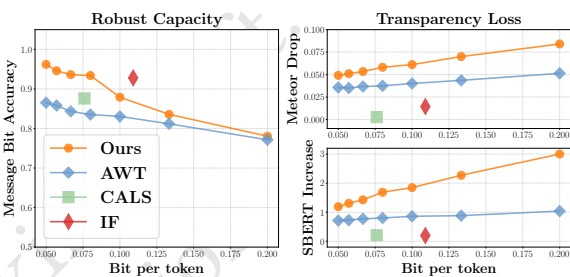

**Figure 4: The robustness performance of different watermarking schemes against DAE. (Our setup: $D_1 = 0$, K=None)**

and $\epsilon = 10^{-9}$ for training. To reduce memory usage, we only retain the top-32 probabilities for each BERT prediction in knowledge distillation. Logit outputs of BERT are passed through a softmax layer with temperature 10. With regard to side information $K$ and distortion constraint $D_{S,X}$, we set $K$ = None and $D_1 = 0$ by default. In particular, $K$ = None ensures a fair comparison between our method and baselines.

The encoder and decoder are adversarially trained against the attacker. Length of each token sequence is sampled with 95% probability from a Gaussian distribution $\mathcal{N}(80, 5^2)$ and 5% probability from $\mathcal{N}(40, 5^2)$, for obtaining sequences of varied lengths. All models are trained on Ubuntu 22.04.2 LTS with a single NVIDIA RTX 3090 GPU. Detailed information about the training time and GPU memory cost is documented in Appendix F.

## 5.2 Performance of Watermarks

We compare our method to baselines in terms of capacity, transparency and robustness.

We evaluate the capacity and transparency under different BPTs in Fig. 3. Note that the bit per token for CALS and IF is not adjustable as the number of bits to be embedded are defined by their position selection algorithms. Hence in Fig. 3 and 4, the performance for CALS and IF is represented by a single point respectively. In terms of capacity, these two rule-based methods reach 100% bit accuracy due to their deterministic algorithms: the prediction of watermark positions as well as the selection of candidate words can be decided without any error, given the watermarked text is not corrupted by adversary. Our method shares a similar capacity performance with

**Table 1: Robustness against adaptive attack, ours vs. baselines. The bit accuracy on corrupted texts represents the robust capacity of the watermarking system, while SBERT increase and meteor drop denote the decline of text quality before and after attacks. The arrows indicate desirable directions. (Ours setup: $D_1 = 0$, K=None)**

| BPT | Bit Acc (corrupted) ↑ | | Meteor Drop ↑ | | SBERT Increase ↑ | |
|---|---|---|---|---|---|---|
| | AWT | Ours | AWT | Ours | AWT | Ours |
| 0.050 | 0.615 | **0.714** | 0.026 | **0.057** | 0.512 | **1.227** |
| 0.057 | 0.624 | **0.702** | 0.026 | **0.059** | 0.487 | **1.300** |
| 0.067 | 0.612 | **0.688** | 0.028 | **0.063** | 0.549 | **1.517** |
| 0.080 | 0.605 | **0.674** | 0.033 | **0.071** | 0.648 | **1.777** |
| 0.100 | 0.623 | **0.654** | 0.029 | **0.070** | 0.560 | **1.820** |
| 0.133 | 0.612 | **0.623** | 0.036 | **0.080** | 0.638 | **2.197** |
| 0.200 | **0.617** | 0.606 | 0.048 | **0.097** | 0.935 | **2.894** |

that of AWT, and is merely 0.028, 0.031 shy of AWT in terms of bit accuracy at 0.13, 0.2 BPT, respectively.

Our transparency described by Meteor score is slightly lower than that of AWT, mostly because the metric directly describes the number of changed tokens, but our method tends to frequently replace the original tokens with synonyms. That would not necessarily result in a worse transparency, as indicated by the SBERT distance where ours largely remains comparable to that of AWT. CALS and IF enjoys better transparency since their infill models always predict candidate words with minimum effect on text semantics.

For robustness, our primary focus is on two categories of attacker models: the denoising auto encoder (DAE) [24] and the adaptive attacker. A thorough description and comparison of the attacks can be found in Appendix C.

In face of DAE attack (Fig. 4), our method achieves a remarkable robustness in terms of robust capacity at greater transparency losses, indicating that not only does our method successfully defend DAE, but also induces the attacker to ruin the usability of the watermarked text. DAE attack is most effective to AWT and CALS as their bit accuracies decline significantly from that of plain capacity. IF, instead, sacrifices transparency to gain a better robustness against DAE, revealing the tradeoff between transparency and robustness to some extent.

The results against adaptive attack in Table 1 also demonstrate the better robustness performance of our model. We omit the rule-based methods, i.e., CALS and IF, from the table since once the detail of the methods are made public, the adversary could easily figure out the position of watermarks and remove them with nearly 100% probability. AWT suffers greatly from the adaptive attack, with bit accuracy as low as 0.6, which is close to random guess (0.5) at all BPTs, meaning that the adaptive attacker beats the watermarking party in removing the watermarks. By contrast, the robust capacity of our method remains over 0.7 when BPT is low. Although the message bit accuracy decreases as BPT gets larger, the transparency loss correspondingly increases, indicating that the text semantics have been severely distorted.

Besides, we summarize the statistics of watermark patterns in Table 2. Compared to AWT, our framework significantly increases

the total number of patterns with an acceptable transparency compromise. Among all components, the knowledge distillation (KD) plays an important role in preserving text naturalness by improving the diversity of patterns while lowering the average occurrence of each pattern below 3, which set barriers for an attacker to learn the mapping from $X$ to $S$ by pattern detection.

**Table 2: Statistics of watermark patterns. Occ. means occurrences.**

| | No. of Patterns | No. of Occ. | Avg Occ./Pattern |
|---|---|---|---|
| Ours | 9671 | 28295 | 2.926 |
| Ours (w/o KD) | 1546 | 24531 | 15.87 |
| AWT | 675 | 9701 | 14.37 |

In sum, our method has achieved the optimal performance with high capacity, strong robustness and acceptable transparency compromises.

## 5.3 Effect of Payoff Function

We investigate whether the theoretical payoff function Eq. (4) truly works as expected in our system by removing the payoff from the training loss of Eq. (12). In that case, the watermarking party merely pits against the attacker by optimizing its message reconstruction losses under distortion constraints. It turns out, as shown in Fig. 5, although our method is slightly inferior to that without the payoff in terms of capacity, our method exceeds by a large margin (around 0.05 bit accuracy) in robust capacity. It clearly indicates that the payoff function plays a vital role in ensuring a robust watermarking scheme. As to the transparency, ours mostly enjoys a higher transparency under attack or not, suggesting that the payoff function not only enhances robustness but also seeks a better tradeoff in capacity and transparency. The results again highlight the importance of principled design.

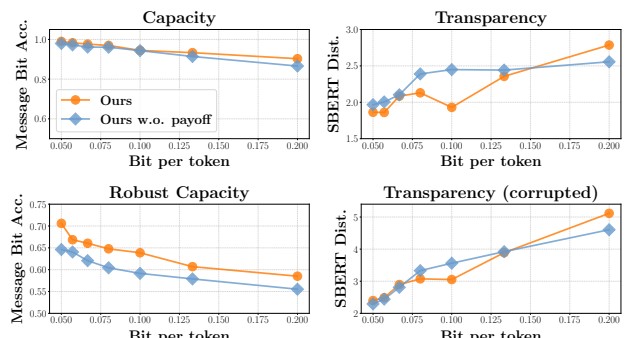

**Figure 5: The capacity, transparency and robustness performance of our framework with & without the payoff function. (Setup: $D_1 = 1$, K=None)**

## 5.4 Effect of Side Information

To investigate how side information affects the performance of watermarks, we set $D_1 = 1$, which provides greater flexibility in

watermark generation, allowing for a better observation of the influence. Moreover, we opted for a simpler MINE network to enhance training efficiency. Details are provided in Appendix A. We consider the following types of side information as $K$.

• *Original text S*. The decoder relies on the presence of the original text for decoding. In implementation, embeddings of $S$ are concatenated with embeddings of $X$ and $T$ before being fed into the decoder.

• *Half of S*. The implementation is similar to $K = S$ except that we use the first half of the embeddings of $S$ as the side information.

• *Token-wise difference between S and X (Diff)*. The scene can be considered as a storage-optimization of $K = S$. The decoder first tries to recover $S$ from the token-wise difference and then decodes as case 1 does.

• *Locations of watermarks (Loc)* are used by the decoder to recover the original text $S$ with the help of a pre-trained masked language model [7]. Masking $X$ or $Y$ on the locations of watermarks, the pre-trained model predicts the tokens on these locations.

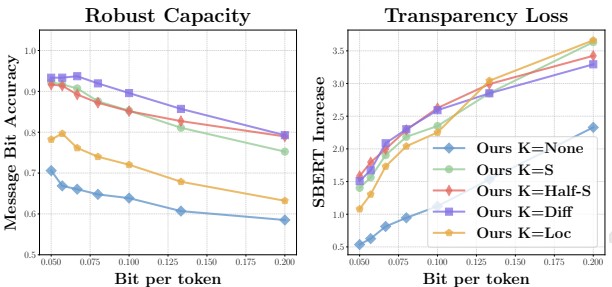

**Figure 6: The capacity, transparency, and robustness of our watermarking system with different forms of side information $K$. (Setup: $D_1 = 1$, robustness is evaluated under adaptive attack.)**

The results are shown in Fig. 6. We find that all forms of side information improve the robust capacity of the watermarking system compared to the case without, i.e., $K =$ None. The reason is that the side information provides additional information for the decoder to recover the original text, which allows messages to embed with less distorted and stealthier patterns.

## 5.5 Ownership Verification

We show how our system works in the actual ownership verification case. Given a text, we first split it into several sequences with fixed length, and embed bits into each sequence. Following AWT [1], we apply hypothesis test to aggregated sequences where 20 bits of watermark are extracted from 5 consecutive sequences. We test the difference between ground-truth messages and the decoded ones. Detailed steps for ownership embedding and verification are introduced in Appendix E.

We set the significance level $\alpha = 0.05$. If the $p$-value is lower than $\alpha$, we can reject the null hypothesis and claim that the text has been watermarked. The success rate of ownership verification is defined as the rate of rejecting the null hypothesis when the text is watermarked. The results are provided in Table 3 with number of sequences $n = 5$ and watermark length $L = 4$. As discussed,

the success rates of rule-based methods achieve 100% under no attack, but drop to 0 under an adaptive attack. Hence the vulnerable cases are omitted. At BPT= 0.05, our method achieves a success rate of 100% and 97.8% in ownership verification, in face of DAE attacks and adaptive attacks, respectively. Across different BPTs, our method maintains a success rate over 94% against DAE attacks, 61% against adaptive attacks, well surpassing the performance of baselines. We also observe that confronted with the strong adaptive attacks, leveraging side information $K = S$ helps retain a high robustness level.

**Table 3: Success rates of ownership verification under different attacks. (Setup: $D_1 = 1$, $K =$ None, $n = 5$, $L = 4$.)**

| BPT | | 0.05 | 0.1 | 0.2 |
|---|---|---|---|---|
| No Attack | AWT | 0.965 | 0.949 | 0.798 |
| | Ours | **1.000** | **0.994** | **0.967** |
| DAE Attack | AWT | 0.887 | 0.817 | 0.660 |
| | CALS | 0.930 (BPT = 0.076) | | |
| | IF | 0.942 (BPT = 0.109) | | |
| | Ours | **1.000** | **0.985** | **0.948** |
| Adaptive Attack | AWT | 0.195 | 0.218 | 0.243 |
| | Ours | 0.463 | 0.264 | 0.140 |
| | Ours K=S | **0.978** | **0.881** | **0.611** |

## 6 CONCLUSION

In this paper, we propose the first principled design for text watermarking founded on the theoretical basis of information-hiding framework. The watermarking party, including an encoder and a decoder, learns the watermarking strategy against a removal attacker channel in a rate-distortion-constrained capacity game, to achieve the maximum rate of reliable transmission, i.e., watermark capacity. Our design is a principled approach against an omniscient attacker who is aware of the watermarking model and strategy, a realistic attack model that most of the current watermarking methods fail to defend. By extensive experiments, we verify that our method is superior to the state-of-the-art in terms of watermarking requirement, i.e., capacity, transparency, and robustness.

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
