# OpenReview forum: "A Principled Approach to Natural Language Watermarking"
_acmmm.org/ACMMM/2024/Conference — MM2024 Poster_

### Official Review · Reviewer_kWht · 2024-05-07

**Rating:** 4
**Confidence:** 2

**Summary:**

This paper proposes a watermarking solution designed specifically for safeguarding the intellectual property rights of natural language models.

**Strengths:**

1. This paper introduces an innovative embedding principle, proposing the first principled design for natural language watermarking grounded in a theoretical information-hiding framework.
2. The technical solution presented is sufficiently innovative.
3. The experiments conducted are comprehensive, and through comparisons with multiple state-of-the-art watermarking systems, the superiority of the proposed scheme in terms of performance has been convincingly demonstrated.

**Limitations:**

1. Since the proposed method in this paper belongs to post-watermarking, it is only effective when users can only access the model's API and not the model itself. The authors should clarify this point in the paper.
2. The authors should provide a visual comparison of the text before and after watermark embedding to demonstrate the differences intuitively.
3. There exists a trade-off between robustness and embedding capacity, where typically, the upper limit of embedding capacity for robust watermarking schemes lacks practical significance, as adjusting the scheme's parameters to achieve this limit often leads to significantly compromised robustness. Therefore, regarding the experiments presented in Figure 5, I am curious about how the authors arrived at the conclusion that, "It turns out, as shown in Fig. 5, although our method is slightly inferior to that without the payoff in terms of capacity, our method exceeds by a large margin (around 0.05 bit accuracy) in robust capacity."
4. Regarding robustness, it is advisable for the authors to discuss in the paper whether the two considered attacks, " the denoising auto encoder (DAE) and the adaptive attacker," are sufficient to cover real-world application scenarios.

**Suitability:**

3

---

### Official Review · Reviewer_nNst · 2024-05-10

**Rating:** 4
**Confidence:** 3

**Summary:**

This paper studies an important and interesting topic: text watermarking. The authors aim to adds watermarks (post-processing watermarks) to existing texts such as natural texts and texts generated by LLM. The watermarking method in this paper is inspired by information hiding technology and implements watermark embedding and extraction based on the extended E-N-D framework, which makes the method theoretical.

**Strengths:**

1. This paper studies an important and interesting topic: text watermarking.
2. The watermarking method and the design of the watermark system have a theoretical basis.
3. The experiments in the paper are generally solid.

**Limitations:**

1.	Although the method in the paper has better performance in terms of robustness, it has a weaker performance than that of CALS and IF in terms of capacity and transparency.
2.	Although the design of the watermarking system has a theoretical basis and is somewhat novel, the E-N-D watermarking framework has been widely used in image watermarking. The authors need to explain more about the novelty of the watermarking system.

Minor Comments:
The formulas in Section 3 are defined in Section 4, which reduces reading fluency.

Question:
1.	What does A in formula (1) mean? Does it represent a channel?
2.	I don’t understand the meaning of the arrows in Table 1. Why are SBERT increase and meteor drop the expected directions? After the watermark text is attacked, shouldn't the decrease in meteor and the increase in SBERT mean that the smaller the better?
3.	What is the message bit accuracy after watermark text is rewrite attacked?

**Suitability:**

2

---

### Official Review · Reviewer_YcBu · 2024-05-24

**Rating:** 4
**Confidence:** 4

**Summary:**

This paper proposes a principled approach to natural language watermarking based on an information-hiding framework. The watermarking party encodes watermarks into the text, and the attacker tries to remove them, with the assumption that the attacker is omniscient and aware of the watermarking strategy. Side information, unknown to the attacker, is incorporated to enhance the system's efficacy and robustness.

**Strengths:**

1. The design of the watermarking system is based on a theoretical information-hiding framework, which provides a solid foundation;
2. The system is designed with the consideration of an omniscient attacker, thereby enhancing its robustness against realistic attacks;
3. The thoughtful inclusion of side information is a strategic move to bolster the system's defenses and efficiency.

**Limitations:**

1. This paper shows that the side information can improve capacity, transparency, and robustness of the watermarking system. However, the system sacrifices public verifiability to achieve robustness against removal attacks, limiting its applicability in certain scenarios.
2. The role of side information is crucial in the process of ownership verification, necessitating additional considerations for its management and distribution;
3. As depicted in Figure 3, the proposed method exhibits a noticeable performance gap when compared to baseline methods. This raises questions about the efficacy of the trade-off between normal performance and robustness;
4. As shown in Appendix F, the training cost of offline stage is computationally expensive. What is the cost of online stage?
5. The robustness of the system against more removal attacks is not evaluated, such as word insertion/substitution/deletion, paraphrase[a], Copy-Paste[b] and Emoji Attack[c];
6. This paper omits comparisons with post-processing methods, such as those referenced in [d], [e], [f], and [h].

[a] Krishna K, Song Y, Karpinska M, et al. Paraphrasing evades detectors of ai-generated text, but retrieval is an effective defense. NeurIPS, 2023.

[b] Kirchenbauer J, Geiping J, Wen Y, et al. On the Reliability of Watermarks for Large Language Models. ICLR. 2024.

[c] Kirchenbauer J, Geiping J, Wen Y, et al. A watermark for large language models. ICML. 2023.

[d] Zhang R, Hussain S S, Neekhara P, et al. Remark-llm: A robust and efficient watermarking framework for generative large language models. USENIX Security. 2024.

[e] Liu A, Pan L, Hu X, et al. A Semantic Invariant Robust Watermark for Large Language Models. ICLR. 2024.

[f] Fu Y, Xiong D, Dong Y. Watermarking conditional text generation for ai detection: Unveiling challenges and a semantic-aware watermark remedy. AAAI. 2024.

[h] Zhao X, Ananth P V, Li L, et al. Provable Robust Watermarking for AI-Generated Text. ICLR. 2024.

**Suitability:**

3

---

### Meta-Review · Area_Chair_XDVF · 2024-07-01

**Recommendation:** Accept (Poster)
**Confidence:** 5

**Metareview:**

The reviewers generally are satisfied with the authors' rebuttal and recognize the contribution of this manuscript.
Thus, it is recommended to accept this paper.
However, the AC also recommends the authors to carefully check the wording/sentences regarding  "simultaneously improving embedding capacity and robustness'' in the entire paper, as suggested by Reviewer kWht.